# Increased Expression of TLR4 in Circulating CD4+T Cells in Patients with Allergic Conjunctivitis and In Vitro Attenuation of Th2 Inflammatory Response by Alpha-MSH

**DOI:** 10.3390/ijms21217861

**Published:** 2020-10-23

**Authors:** Jane E. Nieto, Israel Casanova, Juan Carlos Serna-Ojeda, Enrique O. Graue-Hernández, Guillermo Quintana, Alberto Salazar, María C. Jiménez-Martinez

**Affiliations:** 1Department of Immunology and Research Unit, Institute of Ophthalmology “Conde de Valenciana Foundation”, Ciudad de México 06800, Mexico; janeeyre@ciencias.unam.mx (J.E.N.); israel.casanova@elconde.org (I.C.); guillermo.quintana@institutodeoftalmologia.org (G.Q.); albertosalazar09@gmail.com (A.S.); 2Department of Cornea and Refractive Surgery, Institute of Ophthalmology “Conde de Valenciana Foundation”, Ciudad de México 06800, Mexico; juanc.sernao@gmail.com (J.C.S.-O.); egraueh@gmail.com (E.O.G.-H.); 3Department of Biochemistry, Faculty of Medicine, National Autonomous University of Mexico, Ciudad de México 04510, Mexico

**Keywords:** allergic conjunctivitis, TLR4, alpha-MSH

## Abstract

Ocular allergic diseases are frequently seen in ophthalmological clinical practice. Immunological damage is mediated by a local Th2 inflammatory microenvironment, accompanied by changes in circulating cell subsets, with more effector cells and fewer T regulatory cells (Tregs). This study aimed to evaluate the involvement of toll-like receptor 4 (TLR4) and α-melanocyte stimulating hormone (α-MSH) in the immune regulation associated with perennial allergic conjunctivitis (PAC). We performed an Ag-specific stimulation during 72 h of culturing with or without lipopolysaccharide (LPS) or α-MSH in peripheral blood mononuclear cells (PBMC), analyzing the cell subsets and cytokines induced by the stimuli. We also determined α-MSH in tear samples from healthy donors (HD) or PAC patients. Our findings demonstrate an immunological dysregulation characterized by an increased frequency of CD4+TLR4+ in the PBMC of patients with PAC, compared to HD. Most of these CD4+TLR4+ cells were also CD25+, and when α-MSH was added to the culture, the percentage of CD4+CD25+FoxP3+ increased significantly, while the percentage of CD69+ cells and cytokines IL-4 and IL-6 were significantly decreased. In tears, we found an increased concentration of α-MSH in PAC patients, compared with HD. These findings indicate a novel mechanism involved in controlling ocular allergic diseases, in which α-MSH diminishes the concentration of IL-6 and IL-4, restoring the frequency of Tregs and down-regulating CD4 activation. Moreover, we demonstrated the involvement of CD4+TLR4+ cells as an effector cell subset in ocular allergy.

## 1. Introduction

The prevalence of allergic conjunctivitis (AC) varies from country to country, with rates between 15% and 40% [1,2,3], and the most affected population is pediatric patients [4]. Clinical forms of AC include chronic inflammation of the ocular surface in atopic keratoconjunctivitis (AKC) and vernal keratoconjunctivitis (VKC). In contrast, in mild forms, the inflammation is mainly localized in the conjunctiva; it is persistent in perennial allergic conjunctivitis (PAC), and there are periods in which no damage is induced in seasonal allergic conjunctivitis (SAC) [1,2].

Immunological damage appears to be mediated by the activation of CD4+T cell subsets [5] by environmental and ubiquitous allergens [6,7]. IL-4, IL-5, and IFN-γ are cytokines involved in ocular surface damage in chronic forms of allergy [8]. Interestingly, circulating CD4+T cells in patients with VKC produce IL-4, IL-5, and IFN-γ after *Dermatophagoides pteronyssinus* (*Der p*) stimulation [9]. By contrast, in the acute forms of AC, the cytokines released after *Der p*-stimulation are IL-5, IL-6, and IL-8, and their circulating CD4+T cells express CCR4 and CCR9, which are phenotypes of Th2 cells in transit (potential homing) to the conjunctiva [10]. Sensitization to *Der p* is clinically relevant in ocular allergy [7], and it is one of the most studied allergens in relation to the house dust mite (HDM) [11]. It has protease activity that is capable of disrupting the airway epithelial barrier, initiating the secretion of IL-25, IL-33, and TSLP by damaged cells. These soluble molecules promote Th2 differentiation and recruit eosinophils, basophils, and dendritic cells to the local airway [11,12]. In these models, damage to the epithelium increases the crossing of allergens across the epithelial barrier, favoring the activation of local CD4+T cells and the diminution of Treg activity. Remarkably, patients with PAC have diminished the frequency of circulating CD4+CD25+FoxP3+ (Tregs) cells [10].

The disruption of the epithelial barrier facilitates the recognition of pathogen patterns by innate receptors, such as toll-like receptors (TLR) [13]. TLR-4 has been well recognized in the development of asthma, with an increased expression on the stromal cells in the airways of a mouse model [14]. In line with this, TLR4 is increased in epithelial cells and co-expressed in conjunctiva-infiltrating CD4+T cells in patients with VKC [15]. After TLR4 recognizes lipopolysaccharide (LPS), it induces several intracellular signals leading to the secretion of inflammatory cytokines, such as TNF-α and IL-6 [13]. Notably, in a rat model of endotoxin-induced uveitis, alpha-MSH, a neuropeptide expressed at the ocular level, suppressed LPS-induced inflammation [16], and in mouse-derived macrophages, alpha-MSH was able to suppress the activation of LPS-stimulated TLR4 [17]. These data are relevant because α-MSH has also been linked to the induction of Tregs in mice [18]. Nevertheless, neither the involvement of TLR4/α-MSH in the acute forms of ocular allergy nor their functional status in circulating CD4+T cells have been studied in patients with PAC yet. This work aims to evaluate the potential involvement of TLR4/α-MSH in CD4+T cells and their functional impact in the cells of patients with perennial allergic conjunctivitis.

## 2. Results

### 2.1. Increased Frequency of CD4+TLR4+T Cells in Peripheral Blood Mononuclear Cells in Patients with Perennial Allergic Conjunctivitis

We first investigated the percentage of CD4+TLR4+ cells in the peripheral blood of patients diagnosed with PAC and healthy controls (see Table 1 for demographic data and Appendix A). We found a higher frequency of CD4+TLR4+ T cells in the peripheral blood of patients diagnosed with PAC, compared to healthy donors (HD) (MD 5.6%, IQR 3–39.83.65 versus MD 2.6%, IQR 1.3–4.3, respectively; *p* = 0.0442). As reported previously, when we evaluated the percentage of circulating CD4+CD25+FoxP3+ cells, we observed a 5-fold lower frequency of Tregs in PAC patients, compared with HD (MD 1.04%, IQR 0.395–1.515 versus MD 9.89%, IQR 4.01–10.07, respectively; *p* = 0.0001) (Figure 1). Interestingly, when we analyzed the TLR4 expression in CD4+CD25+ cells, we observed a 4.5-fold higher frequency in PAC patients, compared with HD (MD 12.3%, IQR 3–49 vs. MD 2.7%, IQR 1.8–1.4, respectively; *p* = 0.0042) (Table 2).

Additionally, we observed the differential expression of the mean fluorescence intensity (MFI) in TLR4 and FOXP3 in PAC patients and HD. The TLR4 MFI in CD4+T cells was also higher in PAC patients, compared with HD; however, the FOXP3 MFI in CD4+CD25+T cells was lower in PAC patients than in HD (Table 3).

### 2.2. CD4+TLR4+T Cells Are Induced after Der p Stimulation 

In order to explore whether CD4+TLR4+ cells were induced by Ag-specific stimulation (*Der p*) or by LPS stimulation, we obtained peripheral blood mononuclear cells (PBMC) from patients with perennial allergic conjunctivitis. Then, the PMBCs were cultured and stimulated with *Der p* and/or LPS. After 72 h of culturing, we observed a significant increase in the percentage of CD4+TLR4+ cells when the PBMC were stimulated with *Der p*, compared to when they were stimulated by RPMI-1640 alone (MD 5.4, IQR 2.7–6.2 vs. MD 1.7%, IQR 0.41–2.2, respectively; *p* = 0.0108) (Figure 2). We did not observe significant synergic activity in the induction of CD4+TLR4+ cells after *Der p* + LPS stimulation (MD 7.5, IQR 3.7–9.9) (Table 4). However, when we evaluated the activation of CD4+T cells, we observed a significant expression of CD69 after *Der p* or LPS stimulation, compared to when they were stimulated by RPMI alone (MD 2.9%, IQR 2.2–3.8 versus MD 2.6%, IQR 1.9–5.6, respectively; *p* = 0.0137 vs. *p* = 0.0147, respectively). A significantly higher CD69 expression in CD4 T cells was observed when the cells were incubated with both stimuli (MD 3.8, IQR 1.8–11.3; *p* = 0.041), compared to when they were incubated with *Der p* alone (Table 4). Remarkably, the expression of TLR4 in CD4+CD25+ was significantly increased after stimulation with *Der p* and LPS, compared to when they were stimulated with RPMI alone (*p* = 0.0303 vs. *p* = 0.0173, respectively). We did not observe significant synergic activity in the induction of TLR4 in CD4+CD25+ cells after *Der p* + LPS stimulation (MD 8.2, IQR 7.8–15.3) (Table 4). These results demonstrate that Ag-specific stimulation induces TLR-4 expression in T cells, and it is mainly expressed in CD25+ cells.

### 2.3. α-MSH Induces Treg Cell Differentiation In Vitro from PBMC after Der p Stimulation

We explore the possibility of inducing Treg CD4+CD25+FOXP3+ cells from the PBMC of patients with allergic conjunctivitis after specific allergen stimulation. As expected, we observed a significantly increased frequency of CD4+CD25+FOXP3+ cells after *Der p* stimulation (MD 3.29%, IQR 2.77–3.87), compared with the non-stimulated cells (MD 0.32%, IQR 0.23–1.325) (*p* = 0.0013). No significant increment of Tregs was observed when the PBMC were stimulated with LPS. Nevertheless, the frequency of CD4+CD25+FOXP3+ cells was significantly increased when α-MSH was added to *Der p*-stimulated cells (MD 4.56%, IQR 3.020–5.435) (*p* = 0.0437). No significant increment was observed in the percentage of Tregs with LPS + *Der p* and α-MSH, nor when α-MSH was added alone (Table 4 and Figure 3).

### 2.4. α-MSH Inhibits the Expression of CD69 in CD4+T Cells and Decreases IL-4 and IL-6 after Der p Stimulation

In addition to determining the percentage of CD4+CD25+FOXP3+, we explored the expression of CD69 after *Der p* stimulation plus α-MSH. We observed a 2-fold lower expression of CD69 in CD4+ T cells when they were stimulated with *Der p* and α-MSH (MD 1.61%, IQR 0.557–2.278), compared with *Der p* alone (MD 2.73%, IQR 0.5575–2.278) (*p* = 0.0147) (Figure 4). Then, we analyzed supernatant cytokines and observed that the IL-4 concentration was diminished by 6.7 times (*p* = 0.005) when α-MSH was added. Similarly, IL-6 was diminished by 100 times (*p* = 0.0006) when α-MSH was added to the culture, compared to when it was stimulated by *Der p* alone. On the other hand, stimulation with *Der p* increased the TNF-α concentration by 20 times (*p* = 0.0003). We did not observe significant changes in the TNF-α concentration when α-MSH was added to the culture. Even though we did not observe significant changes in the production of IFN-gamma with *Der p* or *Der p* plus α-MSH, when we analyzed the ratio of IL-4/IFN-gamma, we found that only when α-MSH was added to the culture did we observe a significant change in the IL-4/IFN-γ index (*p* = 0.0313) (Figure 4).

### 2.5. Increased Concentration of α-MSH in the Tears of Patients with Allergic Conjunctivitis

α-MSH was determined in the tears and serum of patients with AC and HD. The results are shown in Figure 5. We observed that α-MSH was increased by 2.1 times in the tears of PAC patients, compared with HD (*p* = 0.0218). Additionally, α-MSH was increased by 2.5 times in the serum of PAC patients, compared with the serum of HD (*p* = 0.0001). Furthermore, we observed a significantly higher α-MSH concentration in tears, compared with the α-MSH concentration, in the serum of both groups (Figure 5).

## 3. Discussion

Conjunctival inflammation in acute forms of ocular allergy has increased CCR4+CCR9+ effector T cells, with a diminished frequency of CD4+CD25+FOXP3+ regulatory T cells in peripheral blood [10]. Bonini et al. [15] reported an infiltration of CD4+TLR4+ cells in the conjunctiva of VKC patients, suggesting a significant role of these cells in the chronic forms of AC. Interestingly, Taylor et al. [17] reported, in the macrophage cell line, J774, and adherent spleen macrophages of mice, that alpha-MSH was able to inhibit LPS-stimulated cells. Thus, this study aimed to evaluate the potential involvement of TLR4/α-MSH in CD4-activated cells in patients with perennial allergic conjunctivitis.

In this work, we observed an increased expression of TLR4 in the CD4 T+ cells of patients with perennial allergic conjunctivitis (PAC). TLR4 is a pattern-recognition receptor (PRR) that recognizes bacterial LPS and the *Der p2* allergen [19]. *Der p 2* is a major allergen found in *Dermatophagoides pteronyssinus* (*Der p*) [20]. Stimulation of TLR4 trough *Der p2* induces IgE secretion and an inflammatory response characterized by eosinophils and lymphocytes in experimental allergic asthma [20]. The function of human CD4+TLR4+T cells has been studied by other authors, showing that LPS stimulation increases cell adhesion to fibronectin by CD4+T cells [21]. Remarkably, when we stimulated PBMC in patients with perennial allergic conjunctivitis with *Der p*, we observed that TLR4 and CD69 were increased in CD4+T cells, suggesting that allergen-specific stimulation also induces TLR4. Remarkably, when we stimulated the PBMC of patients with perennial allergic conjunctivitis with *Der p*, we observed an increased TLR4 and CD69 in CD4+T cells, suggesting that allergen-specific stimulation also induces TLR4. In line with our results, Sahoo et al. showed, in a mouse model, that the neutralization of TLR4 in splenic T cells using a peptide antagonist and stimulating with anti-CD3 and anti-CD28 decreased the CD69 expression, suggesting a role of TLR4 in acute effector responses induced by T cell receptor (TCR) [22]. On the other hand, TLR4 signalization on mice dendritic cells inhibits Treg cell differentiation through IL-6 lacking immune regulation [23,24]. Further studies isolating CD4+TLR4+ cells from blood samples from allergic conjunctivitis patients and stimulating them with *Der p* are needed to determine if CD4+TLR4+ cells have a role in the induction of the IL-6 pro-inflammatory microenvironment and the inhibition of Treg cells.

Treg cell differentiation is directed by the recognition of self-antigen-MHC complexes in the thymus [25] and in the periphery through the cytokines, TGF-β and IL-2, and TCR-Ag-MHC [26]. The implication of other soluble molecules, such as α-melanocyte-stimulating hormone (α-MSH), has been linked to the induction of the Treg differentiation in a mouse model [18]. In this work, we observed that the addition of α-MSH to *Der p*-stimulated cells induced CD4+CD25+FOXP3+ cells and also diminished the CD69 expression in CD4+T cells. CD69 is an early activation marker expressed transiently in lymphocytes, and its function is related to proinflammation, engaging blocking antibodies against CD69, which inhibits the ability of T cells to activate macrophages by contact and diminish the activation of effector T cells [27]. Similar to us, Fang et al. showed that SVα-MSH, an analog of α-MSH, down-regulated the CD69 expression in autoreactive cells and induced CD4+CD25+FOXP3+T cells in a mouse model of autoimmune encephalomyelitis [28]. On the one hand, in a mouse model of allergic airway inflammation, α-MSH decreases the production of IL-4 and IL-13 from a bronchoalveolar lavage [29]. In this work, we observed the down-regulation of IL-4 and IL-6 when α-MSH was added to cells stimulated with *Der p*. Our results suggest that α-MSH induces Treg-differentiation and also diminishes the Th2 inflammatory microenvironment. We did not observe IL-10 or TGF-b secretion, supporting that α-MSH suppression could be cell-dependent and lead to the differentiation of effector or regulatory activity. To understand the differentiation of Treg cells induced by α-MSH in allergic conjunctivitis, cAMP, CREB, and ERK need to be explored, which will lead to the uncovering of the mechanism involved in the regulatory activity induced by α-MSH.

Although we used high LPS doses, we observed effector Th2 responses in cultured cells from PAC patients. High LPS concentrations induce an IL-10 enriched microenvironment [30]; thus, once differentiated CD4+TLR4+ cells are committed with effector functions activating a Th2 inflammatory response independently of LPS concentration. Similarly, the *a*-MSH dose evaluated in this work was not at physiological concentration but was able to down-regulate the activation of effector cells. Hence, the results presented here show a biological phenomenon observed in cells isolated from allergic patients that could be used as a starting point to evaluate the pharmacological development of immune-modulators based in *a*-MSH to regulate the allergic process.

α-MSH is a neuropeptide produced by the hypothalamus, keratinocytes, and lachrymal glands [31]. At the ocular level, α-MSH is involved in the homeostasis of the ocular surface in a mouse model of dry eye [32]. Previous studies have demonstrated higher concentrations of neuropeptides in ocular allergic reactions, including substance P (SP), calcitonin gen-related peptide (CGRP), and vasoactive intestinal peptide (VIP), after allergenic challenges, suggesting their participation as inflammatory mediators. Nevertheless, the involvement of α-MSH has not been fully explored in relation to allergic conjunctivitis. Kleiner et al. showed that the basophils from allergic rhinitis patients decreased CD203 after stimulation with α-MSH basophils and allergens or anti-IgE, suggesting that α-MSH suppressed the proinflammatory effector function in human basophils [33]. In our study, PAC patients showed an increased α-MSH concentration in tears and serum, compared with healthy controls. It is well known that proinflammatory cytokines induce α-MSH as a neuroendocrine axis in order to down-regulate inflammation [31]. Thus, it would be necessary to determine the expression of the melanocortin receptors in T cells and ocular surface cells, since it is possible that these cells may not respond to α-MSH or lack immune regulation, despite the fact that α-MSH is increased.

Schirmer’s test without anesthesia measures both basal and reflex tearing, and tear break up time measures the tear film stability of the ocular surface, thus providing a functional measure of mucin, aqueous, and lipid layer. Tearing, itching, foreign body sensation, and photophobia are all common symptoms of allergic conjunctivitis in which inflammation and mechanical irritation alter the density and function of goblet cells and accessory lacrimal glands. Long-term consequences of allergic conjunctivitis may result in fibrotic changes in the ocular surface, thus significantly altering Schirmer´s test and TBUT. However, in the acute stages, Schirmer´s test may be greater than expected due to increased sensibility of the ocular surface, and TBUT may also be altered even in young allergic patients [34]. Although mostly used in dry eye syndrome, both tests are clinically relevant to explore the function of the ocular surface unit, together with the clinical examination under the biomicroscope. Meibomian glands produce the lipids that form the surface lipid layer [35], and secretion of meibum could be influenced by the melanocortin 5 receptor (MCR5) expressed on gland cells [36,37]. Furthermore, α-MSH down-regulates MUC5AC in cultured epithelial cells from the nasal mucosa through TNF-*a* inhibition [38]; thus, anti-inflammatory actions of *a*-MSH could be contributing to change the tear stability. Whether tear α-MSH is a local compensation mechanism to increase the meibomian gland’s function or is a neuroendocrine-immune altered pathway contributing to changes in the mucin secretion at the ocular surface is not known and needs further investigation.

In sum, the activation of CD4+TLR4+T cells by the antigen and TLR induces an inflammatory microenvironment characterized by TNF-α, IL-6, and IL-4, which favors a lack of immune regulation in perennial allergic conjunctivitis. Interestingly, the addition of α-MSH diminishes IL-6 and IL-4, thus down-regulating CD4 activation and restoring the frequency of Tregs. 

## 4. Materials and Methods

### 4.1. Patients and Healthy Donors

Fourteen patients with perennial allergic conjunctivitis and nine healthy donors were enrolled in this study. The diagnosis of active perennial allergic conjunctivitis was based on clinical ophthalmological and allergo/immunological examination. This work was approved by the Scientific (CC-034-2011, 28 Nov. 2011), Biosafety (CB-034-2011, 9 Dec. 2011), and Bioethics (CEI-034-2011, 1 Dec. 2011) Institutional Committees at the Institute of Ophthalmology, “Foundation Conde de Valenciana” Mexico City. This work adhered to the Declaration of Helsinki and the E11 Statements of the International Conference of Harmonisation (E11-ICH). All subjects included in this study gave their informed assent for blood and tear sampling after written information was provided.

### 4.2. Clinical Evaluation

The allergic condition was confirmed using a skin-prick test (SPT). Briefly, the outer epithelial layer of the forearm skin was scratched with a lancet without inducing bleeding to avoid false-positive results. Then, the allergen (*Dermatophagoides pteronyssinus, Der p*) or the control (histamine) were instilled over the skin. The SPT was read between 15–20 min after the allergen skin challenge; positive results were considered for *Der p* (wheal ≥ 3 mm diameter) compared with the histamine control. In addition, the serum total IgE (tIgE) was determined in all subjects included in the study.

The ophthalmological investigation was systematically documented by biomicroscopy, according to [39]. Schirmer test was performed without instillation of topical anesthetic, folding a Schirmer paper strip (AMCON Tear Flow Test Strips Nomax, Inc, MO, USA) over the temporal one-third of the lower lid margin inserted in the conjunctival sac for 5 min to measure the production of tears in millimeters. The tear break-up time (TBUT) was evaluated observing the cornea under a slit-lamp (Carl Zeiss, Meditec Inc. CA, USA) with a cobalt blue filter. TBUT was considered as the time required for the appearance of the first break in the tear film after blinking with fluorescein staining (Bio Glo Fluorescein Ophthalmic Strips USP, Gujarat, India), recording the mean value of three measurements.

### 4.3. Monoclonal Antibodies and Reagents

Allophycocyanin (APC)-labeled mouse or phycoerythrin Cy-5 tandem conjugate (PE-Cy5) monoclonal antibodies (mAbs) against human CD4, phycoerythrin (PE) mAbs anti-human CD25, fluorescein isothiocyanate conjugate (FITC) mAbs anti-human CD69, BD™ CompBeads, BD FACS™ Lysing Solution BD Cytofix/Cytoperm™ solution, and BD FACSFlow™ Sheat Fluid were purchased from BD Biosciences (CA, USA). Allophycocyanin Cy-7 tandem conjugate (APC Cy7), APC allophycocyanin conjugate (APC) mAbs anti-human TLR4 and fluorescein isothiocyanate conjugate (FITC) mAbs anti-human FOXP3 were purchased from BioLegend (CA, USA). The RPMI-1640 culture medium, Lipopolysaccharide (LPS) from Escherichia coli 0111: B4, Concanavalin A (Con A) from *Canavalia ensiformis,* and salts were purchased from Sigma Chemical Co. (MO, USA). Fetal calf serum was obtained from HyClone Labs (UT, USA). Sodium pyruvate, L-glutamine, and 2-mercaptoethanol were purchased from Gibco BRL (MD, USA). *Dermatophagoides pteronyssinus* (*Der p*) and histamine control were purchased from Allerstand Co. (Mexico City, Mexico).

### 4.4. Immunofluorescence of Cellular Surface Markers

Four-color staining was performed on the peripheral blood cells using direct immunofluorescence and either AmCyan- or PeCy5-labeled mAbs against CD4, FITC-labeled mAbs against CD69, or PE-labeled mAbs against CD25 and APC-mAb anti-TLR-4. Briefly, 20 µL of whole peripheral blood was incubated with fluorochrome-labeled mAb for 30 min at 25 °C in darkness. After incubation, the red blood cells were lysed with a BD FACS™ Lysing Solution (CA, USA), according to the manufacturer’s instructions. Then, the cells were washed twice with BD FACSFlow™ Sheat Fluid (CA, USA) and processed for intracellular staining with FITC-labelled anti-human FOXP3 antibody, BioLegend (CA, USA).

### 4.5. Immunofluorescence of Intracellular Markers 

After extracellular staining was performed, the cells were fixed and permeabilized with a BD Cytofix/Cytoperm™ solution, according to the manufacturer´s instructions. Then, the cells were incubated with FITC-labeled anti-human FOXP3 antibody and immediately acquired by flow cytometry. In all cases, fluorescence minus one (FMO) controls and anti-mouse Ig and κ/Negative control compensation particle sets (BD™ CompBeads, BD Biosciences, CA, USA) were used. All samples were immediately acquired after immunofluorescence staining.

### 4.6. Cell Cultures

Whole heparinized peripheral blood was diluted 1:2 (vol/vol) in an RPMI 1640 medium (Sigma Aldrich, Darmstadt, Germany). Peripheral blood mononuclear cells (PBMC) were separated on a Ficoll (Eppendorf, Hamburgo, Germany) density gradient by centrifugation at 300 g for 30 min at room temperature. Then, the cells in the interface were collected, washed twice, and counted using a handheld automated cell counter (Millipore Co., MA, USA). The viability was assessed by trypan blue dye exclusion. The PBMC were cultured in 96-well flat-bottomed cell culture plates (Costar, MA, USA) at 2 × 105 cells/well in an RPMI-1640 medium supplemented with one mM sodium pyruvate and two mM L-glutamines and incubated at 37 °C in a 5% CO2 humidified chamber. After 24 h, the culture medium was removed, and a fresh culture medium supplemented with 10% heat-inactivated fetal calf serum and/or *Der p* (0.5 μg/mL), LPS (10 μg/mL), or α-MSH (1 μg/mL) was added. The Con A mitogen (1 μg/mL) was used as a cell stimulation positive control (Appendix A). After 72 h of culturing, the cells were harvested and processed to measure the intracellular FOXP3 expression and the expression of CD24, CD69, and TLR4 on the cell surface by flow cytometry. The supernatants were collected and stored at −70 °C to determine soluble cytokines. 

### 4.7. Flow Cytometric Analysis

All cells were analyzed for the expression of phenotypic markers in a BD FACSVerse™ (BD Biosciences, CA, USA) flow cytometer using the FACSuite Software, version 1.0.5.3841 (BD Biosciences, CA, USA), and 10,000 events were counted. To analyze the staining of the cell-surface markers, a gate was drawn according to the physical properties (forward scatter, FSC; and side scatter, SSC), which corresponded to lymphocytes. Subsequently, a second gate was drawn according to single cells (FSC-H-forward height vs. FSC-A-forward area). Then, CD4+ cells were selected in a third FSC-CD4 dot. To determine the subsets of CD4+T cells, a new plot was obtained showing CD4+ vs. CD25+, CD69+, or TLR-4+ cells. The data are presented as dot plots to analyze the frequency of positive cells, and histograms were drawn to analyze the mean fluorescence intensity (MFI) of FOXP3+ or TLR4+ cells. Control staining was performed using fluorescence minus one (FMO) and BD™ CompBeads (BD Biosciences, CA, USA).

### 4.8. Quantification of Soluble Cytokines in Culture Supernatants

IFN-γ, TNF-α, IL-2, IL-4, IL-6, IL-10, IL-17 (Human Th1, Th2, Th17 cytokine kit), and TGF-b1 (Human TGF-b1 Single Plex Flex Set) (BD Biosciences, CA, USA) were measured with cytometric bead arrays (CBA) in supernatant samples, according to the manufacturer’s instructions (BD Biosciences, CA, USA). The analyte concentrations were calculated by interpolation using standard curves in BD FCAP Array, software version 3.0 (BD Biosciences, CA, USA). The ranges of detection (pg/mL) were: IFN-γ 0–5592, TNF-α 0–5360, IL-2 0–5083, IL-4 0–5068, IL-6 0–5417, IL-10 0–5256, IL-17 9.3–4387, and TGF-*b* 5.2–1531. These values were in agreement with those that have been reported by the manufacturer.

### 4.9. Determination of Total IgE (tIgE)

The serum samples were processed to determine IgE by fluoro-enzyme-immunoassay (FEIA) in an ImmunoCAP Phadia^®^ 100 Laboratory System (Thermo Fisher Scientific Inc., MI, USA), following the manufacturer’s instructions. The results were analyzed by the ImmunoCAP^®^ software, v.4.13 (Thermo Fisher Scientific Inc., MI, USA), and the limit of detection was < 2 kUI/L.

### 4.10. Tear Samples

Tear samples from healthy and allergic eyes were obtained from the conjunctival fornix by adding of 20 µL of a BSS™ sterile saline solution (Alcon Laboratories, Inc., TX, USA), according to the instructions given in [40]. Then, the ocular wash was immediately recovered and stored at −20 °C for cytometric analysis.

### 4.11. Quantification of α-MSH in Serum and Tears

Serum and tear samples were processed to determine α-MSH in a standard sandwich enzyme-linked immune-sorbent assay technology, following the manufacturer’s instructions (AMSH ELISA KIT Aviva systems biology, CA, USA). The limit of detection was 0.6 ng/mL.

### 4.12. Statistical Analyses

A Wilcoxon rank signed test was used to detect significant differences. The analysis was performed using the GraphPad Prism software, v.8.0 (CA, USA). Differences were considered statistically significant when the test yielded *p* values less than 0.05. 

## 5. Conclusions

Our results suggest a novel mechanism involved in controlling ocular allergic diseases in which α-MSH diminishes the concentration of IL-6 and IL-4, restoring the frequency of Tregs and down-regulating the CD4 activation. We also demonstrated the involvement of CD4+TLR4+ cells as an effector cell subset in ocular allergy.

## Figures and Tables

**Figure 1 ijms-21-07861-f001:**
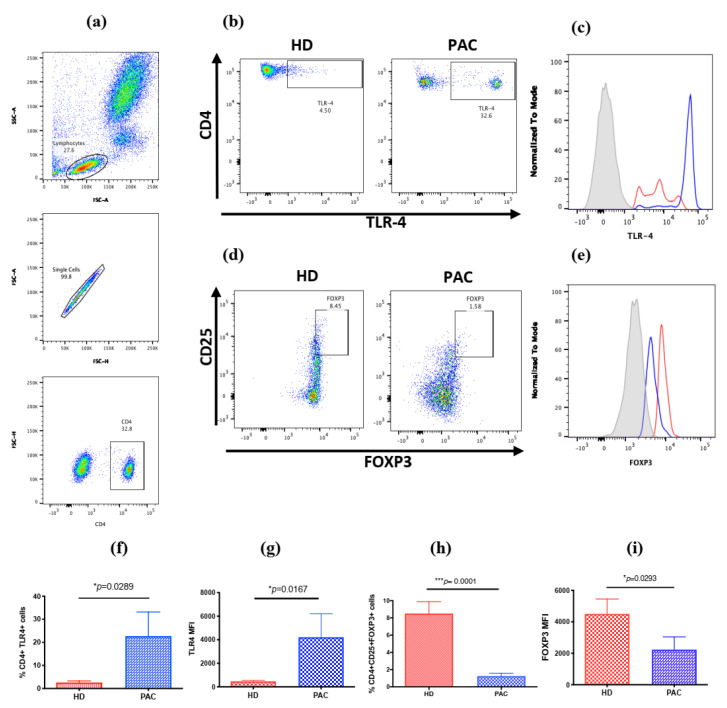
TLR4 expression in CD4+T cells and CD4+CD25+FOXP3+T reg cell subset. (**a**) Both subsets of cells were analyzed from the lymphocyte population through the selection of singlets and gathering of CD4+T cells; (**b**) representative dot plots, showing the frequency of CD4+TLR4+T cells in peripheral blood from patients diagnosed with PAC, compared to healthy donors (HD); (**c**) the TLR-4 expression in CD4 T cells is higher in PAC patients (blue line) than in HD (red line); (**d**) representative dot plots showing the frequency of CD4+CD25+FoxP3+ in PAC patients, compared with that in HD; (**e**) the FOXP3 expression in CD4 T cells is lower in PAC patients (blue line) than in HD (red line); (**f**) percentage of CD4+TLR4+ cells in the peripheral blood of HD and PAC patients; (**g**) the TLR4 expression in the CD4+T cells of HD and PAC patients; (**h**) the percentage of CD4+CD25+FOXP3+ cells in the peripheral blood mononuclear (PBMC) in HD, compared to that in PAC patients; (**i**) the FOXP3 expression in the CD4+CD25+ cells in the peripheral blood of HD and PAC patients. MFI, mean fluorescence intensity. Cytometry figures performed in FlowJo software version 10 (BD Biosences, CA, USA) * *p* ≤ 0.05; *** *p* ≤ 0.001.

**Figure 2 ijms-21-07861-f002:**
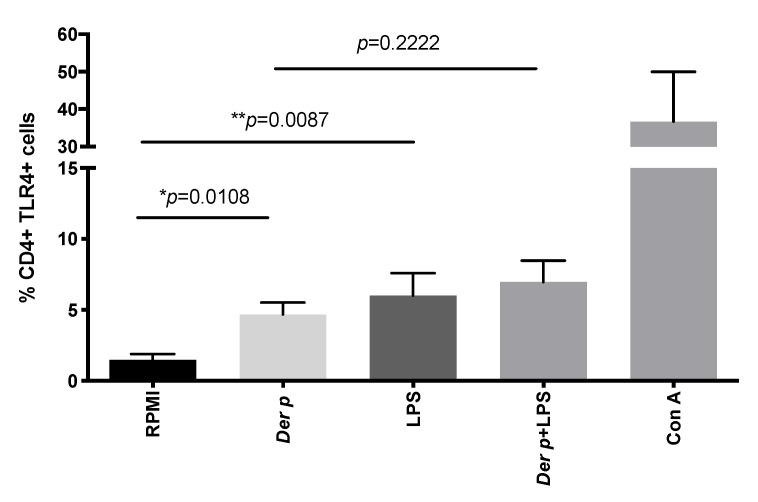
TLR4 expression in CD4+T cells. After 72 h of culturing with *Der p* and LPS, we observed an increased percentage of CD4+TLR4+ cells. * *p* ≤ 0.05; ** *p* ≤ 0.01.

**Figure 3 ijms-21-07861-f003:**
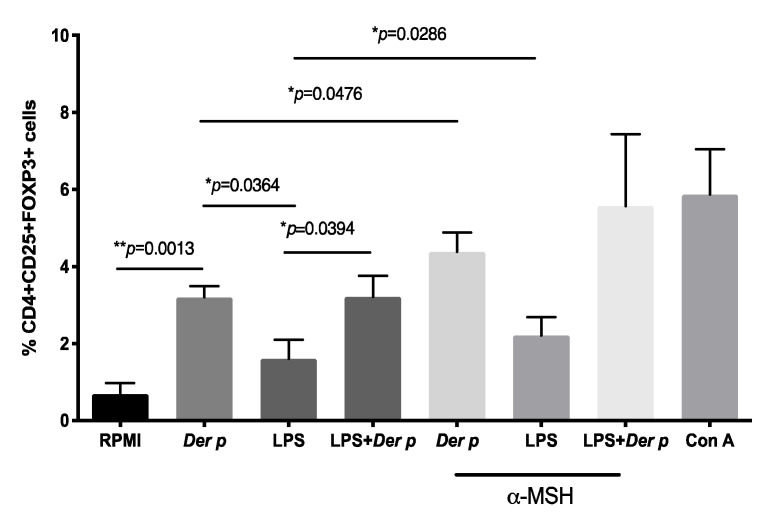
Frequency of T regulatory CD4+CD25+FOXP3+ cells in CD4+T cells after 72 h of culturing with *Der p* and LPS. See Appendix A for α-MSH dose-response experiments. * *p* ≤ 0.05; ** *p* ≤ 0.01.

**Figure 4 ijms-21-07861-f004:**
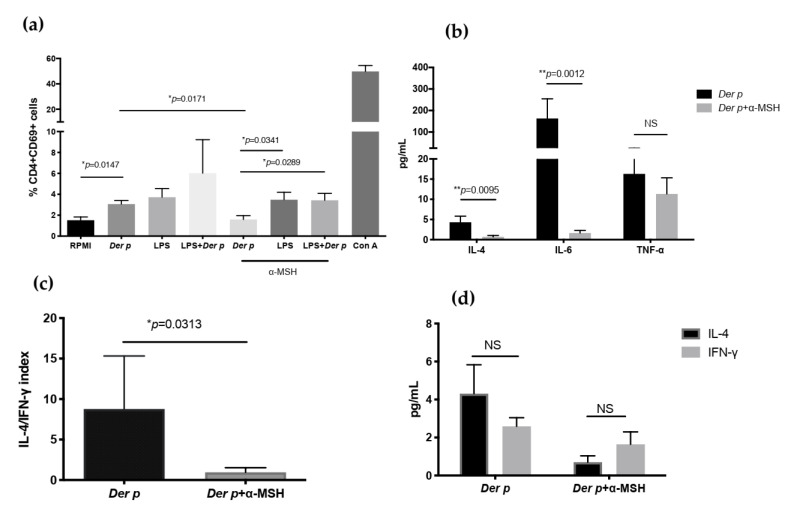
Expression of CD69 in CD4+T cells and decreases in IL-4 and IL-6 after *Der p* stimulation. (**a**) CD69 expression in CD4+T cells after 72 h of culturing with *Der p*, LPS, and α-MSH; (**b**) IL-4, IL-6, and TNF-α in supernatant after 72 h of culturing with *Der p*, LPS, and α-MSH; (**c**) IL-4/IFN-γ index; (**d**) IL-4 and IFN-γ comparison after culturing with *Der p* and α-MSH. * *p* ≤ 0.05; ** *p* ≤ 0.01; NS *p* ≥ 0.05.

**Figure 5 ijms-21-07861-f005:**
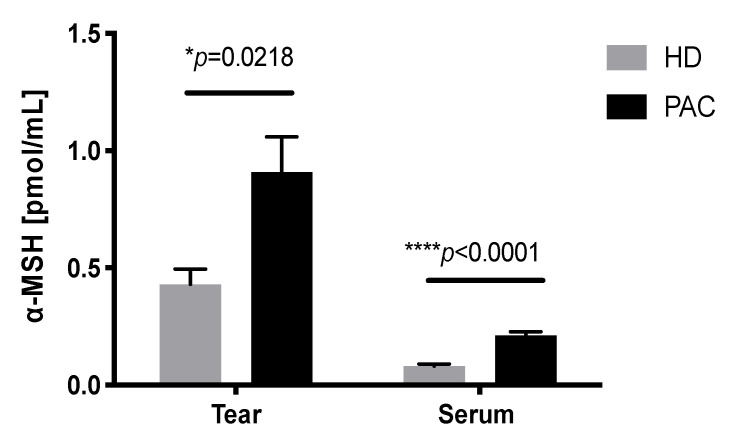
Concentration of α-MSH in the tears of patients with allergic conjunctivitis. α-MSH was measured through an ELISA sandwich of tear and serum samples from healthy donors (HD) and perennial allergic conjunctivitis (PAC) patients. * *p* ≤ 0.05; **** *p* ≤ 0.0001.

**Table 1 ijms-21-07861-t001:** Demographic characteristics of the subjects included in this study.

Demographic Characteristics	HD (*n* = 9) MD (IQR)	PAC (*n* = 14) MD (IQR)	*p* Value
Age	13 (12–14.5)	11 (10–12)	NS
Male	14 (12–17.5)	11.2 (9–12.75)	NS
Female	16.5 (13.7–19.2)	11 (10–12)	NS
TBUT (s)	7.5 (6–10)	5 (4–6)	* 0.0010
Schirmer Test (mm)	25.8 (20–35)	21 (11.25–28)	NS
SPT (mm) to *Der p*	2.7 (0–7.5)	7.3 (5–9)	** 0.0276
tIgE	49 (19.2–59)	612.3 (842.8–299.8)	<0.0001

HD, healthy donor; PAC, perennial allergic conjunctivitis patients; MD, median; IQR, interquartile rank; NS, non-significant; TBUT, tear break-up time; SPT, skin-prick test; *Der p*, *Dermatophagoides pteronyssinus;* tIgE, total IgE; s, seconds; mm, millimeters (see Appendix A for a detailed description of the characteristics of the demographic data). * *p* ≤ 0.05; ** *p* ≤ 0.01.

**Table 2 ijms-21-07861-t002:** Percentages of the T cell subsets in both study groups.

T Cells Subsets %	HD (*n* = 9) MD (IQR)	PAC (*n* = 14) MD (IQR)	*p* Value
CD4+	33.45 (26.35–39.2)	35.65 (27.85–38.8)	NS
CD4+TLR4+	2.6 (1.3–4.3)	5.6 (3–39.83)	* 0.0289
CD4+CD25+	10.27 (9.31–14.02)	16.7 (12.95–23.15)	** 0.0042
CD4+CD25+TLR4+	2.4 (1.8–3.4)	22.65 (3–49)	* 0.0218
CD4+CD25-TLR4+	2.12 (0.7–3.4)	7 (1.8–31.2)	NS
CD4+CD25+FOXP3+	9.9 (4–10)	1 (0.39–1.5)	*** 0.0001

HD, healthy donors; PAC, perennial allergic conjunctivitis patients; MD, median; IQR, interquartile rank; NS, non-significant. * *p* ≤ 0.05; ** *p* ≤ 0.01; *** *p* ≤ 0.001.

**Table 3 ijms-21-07861-t003:** TLR4 expression in CD4+T cells and FOXP3 expression in CD4+CD25+T cells.

T Cells Subsets MFI	HD (*n* = 9) MD (IQR)	PAC (*n* = 14) MD (IQR)	*p* Value
TLR4	412 (349–566)	491 (356–10524.75)	* 0.0167
FOXP3	5333 (1804–6435)	1229 (831–1877)	* 0.0143

MFI, mean fluorescence intensity; HD, healthy donor; PAC, perennial allergic conjunctivitis patients; MD, median; IQR, interquartile rank. * *p* ≤ 0.05.

**Table 4 ijms-21-07861-t004:** CD4+T cell populations after 72 h of culturing with *Der p*, LPS, and α-MSH.

T Cells Subsets%	RPMI MD (IQR)	*Der p* MD (IQR)	LPS MD (IQR)	*Der p* + LPS MD (IQR)	α-MSH MD (IQR)	*Der p* + α-MSHMD (IQR)	LPS + α-MSH MD (IQR)	*Der p* + LPS + α-MSH MD (IQR)	Con A MD (IQR)	*P* Value
CD4+CD69+	1.35 (0.76–2.4)	2.9 (2.2–3.8)	2.6 (1.9–5.6)	3.8 (1.8–11.3	1.68 (0.52–2.9)	1.6 (0.5–2.3)	3–045 (1.98–4.68)	4 (1.8–11.3)	50.6 (37.8–64.3)	^a^ ** *p* = 0.0057, ^b^ * *p* = 0.0147, ^c^ ** *p* = 0.0056
CD4+TLR4+	1.7 (0.4–2.24)	5.4 (2.7–6.2)	4.5 (3.2–9.6)	7.5 (3.7–9.9)	2 (1.3–3.4)	4 (2–5.3)	1.4 (0.6–3.5)	6.3 (2–10.7)	35.7 (12.3–62)	^a^ * *p* = 0.0108, ^b^ ** *p* = 0.0087, ^c^ *p* = 0.5238
CD4+CD25+	5.18 (2.8–5.7)	10.57 (5.9–14.68)	9.2 (7.7–12.9)	8.22 (7.6–14.35)	5.15 (4–6.2)	9.18 (7.3–12)	8 (5.9–10.4)	14.85 (13.5–16.3)	53 (27.8–62.5)	^a^ ** *p* = 0.0059, ^b^ *** *p* = 0.0003, ^c^ *p* = 0.9038
CD4+ CD25+ TLR4+	0.05 (0.02–0.35)	1 (0.15–2.4)	10.7 (8–13)	8.2 (7.8–15.3)	0.06 (0.01–0.51)	0.54 (0.23–1.5)	0.14 (0.06–0.6)	1.4 (1.06–1.8)	44 (3.3–59)	^a^*p* = 0.0303, ^b^ * *p* = 0.0173, ^c^ *p* = 0.668
CD4+CD25-TLR4+	2.3 (1.2–5.6)	5.8 (4.8–12.8)	4 (2.1–6)	4.2 (3.1–9.6)	5.8 (1.8–10.27)	4.2 (3.1–9.6)	2.8 (1.7–4.1)	1.4 (0.82–7.7)	6.6 (4.2–7.6)	^a^ * *p* = 0.0325, ^b^ *p* = 0.5281, ^c^ *p* = 0.3333
CD4+CD25+FOXP3+	0.32 (0.23–1.32)	3.3 (2.7–3.9)	1.1 (1.05–2.6)	3.75 (1.6–4.3)	0.5 (0.42–2.09)	4.6 (3.02–5.4)	2.08 (0.97–3.5)	4.8 (2.4–9.4)	6.7 (3.4–7.5)	^a^ ** *p* = 0.0013, ^b^ *p* = 0.0958, ^c^ * *p* = 0.0478

^a^, *p* = RPMI vs. *Der p*; ^b^, *p* = RPMI vs. LPS; ^c^, *p* = *Der p* vs. *Der p* +α-MSH. * *p* ≤ 0.05; ** *p* ≤ 0.01; *** *p* ≤ 0.001.

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
