# Peer review of "Increased Expression of TLR4 in Circulating CD4+T Cells in Patients with Allergic Conjunctivitis and In Vitro Attenuation of Th2 Inflammatory Response by Alpha-MSH"

_ijms, 2020, doi:10.3390/ijms21217861_

Round 1
Reviewer 1 Report
This manuscript described an interesting study to evaluate the involvement of TLR4 and α-MSH in the immune regulation associated with perennial allergic conjunctivitis (PAC) using in vivo and in vitro studies with PBMCs and tear. Overall, the research design is logical, experimental methods are appropriate, and the most data are convincing. I have some concerns about the methods and data explanation in this manuscript.
- The concentrations of LPS (10 µg/ml) and α-MSH (1.5 µg/ml) used in PBMC cultures were very high. LPS is about 10-100 times higher than regular doses 0.1-1.0 µg/ml, and α-MSH is 7,500-30,000 times higher than physiological levels in human blood serum (45-200 pg/ml) [1,2]. Why use such high doses to treat the PBMCs? Did you perform the dose-response experiments?
- The “Attenuation of Th2 Inflammatory Response by alpha MSH” in the title is not appropriate. The supportive data are only from in vitro culture of PBMCs treated with α-MSH at huge high levels, which diminishes the concentration of IL-6 and IL-4, and restores the frequency of Treg cells. Authors did not perform the in vivo study to treat PCA patients with α-MSH. Furthermore, it is controversial in PCA patients who had the high Th2 inflammatory responses accompanied by higher a-MSH in serum and tear.
- Please describe full name for an abbreviation “MSH” in Abstract, and correct many a-MSH to α-MSH.
References:
1 Mao, Z. et al. Serum alpha-melanocyte-stimulating hormone may act as a protective biomarker for non-traumatic osteonecrosis of the femoral head. Ann Clin Biochem 55, 453-460, doi:10.1177/0004563217738802 (2018).
2 Du, X. et al. Dynamic changes of alpha-melanocyte-stimulating hormone levels in the serum of patients with craniocerebral trauma. Exp Ther Med 14, 2511-2516, doi:10.3892/etm.2017.4793 (2017).
Author Response
This manuscript described an interesting study to evaluate the involvement of TLR4 and α-MSH in the immune regulation associated with perennial allergic conjunctivitis (PAC) using in vivo and in vitro studies with PBMCs and tear. Overall, the research design is logical, experimental methods are appropriate, and the most data are convincing. I have some concerns about the methods and data explanation in this manuscript.
- The concentrations of LPS (10 µg/ml) and α-MSH (1.5 µg/ml) used in PBMC cultures were very high. LPS is about 10-100 times higher than regular doses 0.1-1.0 µg/ml, and α-MSH is 7,500-30,000 times higher than physiological levels in human blood serum (45-200 pg/ml) [1,2]. Why use such high doses to treat the PBMCs? Did you perform the dose-response experiments?
In our study, we used high doses of LPS and α-MSH in PBMC cultures. It has been reported that LPS activates T cells in a dose-dependent concentration, and high doses of LPS induce Tr 1 responses (Xu H, 2008). Similar to Xu et al., we recently reported that LPS (5-10 mg) promotes IL-10 production in B-isolated cells (Salazar A, 2019). Despite high LPS doses, we observed effector responses, possible through TLR4, suggesting that CD4+TLR4+ cells are committed with effector functions activating a Th2 inflammatory response due to the cytokines found at the supernatant of cultures. Thus, it is possible that other cell subsets or soluble mediators are needed to restore the immunological balance, and one of them could be α-MSH.
a) Xu, H., Liew, L. N., Kuo, I. C., Huang, C. H., Goh, D. L. M., & Chua, K. Y. The modulatory effects of lipopolysaccharide‐stimulated B cells on differential T‐cell polarization. Immunology, 2008, 125(2), 218-228.
b) Salazar A, Casanova-Méndez I, Pacheco-Quito M, Velázquez-Soto H, Ayala-Balboa J, Graue-Hernández EO, Serafín-López J, Jiménez-Martínez MC. Low Expression of IL-10 in Circulating Bregs and Inverted IL-10/TNF-α Ratio in Tears of Patients with Perennial Allergic Conjunctivitis: A Preliminary Study. Int J Mol Sci. 2019 Feb 27;20(5):1035. doi: 10.3390/ijms20051035. PMID: 30818819; PMCID: PMC6429471.
Doses-response experiments were performed using the dose reported by Ru et al. (Ru Y, 2015) in the dry eye animal model. To determine the optimum dose of α-MSH, we obtained blood samples from healthy donors (n=3). PBMC were isolated, and cultured in 96-well flat-bottomed cell culture plates (Costar, Cambridge, MA, USA) pretreated with anti-CD3 (BioLegend, San Diego, CA, USA) at 2 × 105 cells/well in RPMI-1640 medium supplemented with 1 mM sodium pyruvate, 2 mM L-glutamine, and incubated at 37°C in a 5% CO2 humidified chamber After 24 h, the culture medium was removed, and a fresh culture medium supplemented with 10% heat-inactivated fetal calf serum and α-MSH at 1mg/mL or 10 mg/mL was added (Time 0, T0). After 72 h, the cells were harvested and processed to evaluate T cell activation with CD69 expression on CD4+ T cells by flow cytometry. The results related to the doses tested (a-MSH) were added as “Supplemental figure 1”. Page 6, lines 194.
c) Ru Y, Huang Y, et al. α-Melanocyte-stimulating hormone ameliorates ocular surface dysfunctions and lesions in a scopolamine-induced dry eye model via PKA-CREB and MEK-Erk pathways.Scientific reports, 2015, 5(1), 1-14.
In the new version of the manuscript, we discussed high LPS and α-MSH doses used in this work. Page 9, lines 280-288. Also, we corrected the reported dose in methods Page 11, line 400; and the reference section was actualized to include the new bibliography added to the text.
- The “Attenuation of Th2 Inflammatory Response by alpha MSH” in the title is not appropriate. The supportive data are only from in vitro culture of PBMCs treated with α-MSH at huge high levels, which diminishes the concentration of IL-6 and IL-4, and restores the frequency of Treg cells. Authors did not perform the in vivo study to treat PCA patients with α-MSH. Furthermore, it is controversial in PCA patients who had the high Th2 inflammatory responses accompanied by higher a-MSH in serum and tear.
We adapted the title according to reviewer suggestion. In the new version of the paper the title is “Increased Expression of TLR4 in circulating CD4+T Cells in Patients with Allergic Conjunctivitis and in vitro Attenuation of Th2 Inflammatory Response by alpha-MSH” Page 1, Lines 3-4.
- Please describe full name for an abbreviation “MSH” in Abstract, and correct many a-MSH to α-MSH.
References:
1 Mao, Z. et al. Serum alpha-melanocyte-stimulating hormone may act as a protective biomarker for non-traumatic osteonecrosis of the femoral head. Ann Clin Biochem 55, 453-460, doi:10.1177/0004563217738802 (2018).
2 Du, X. et al. Dynamic changes of alpha-melanocyte-stimulating hormone levels in the serum of patients with craniocerebral trauma. Exp Ther Med 14, 2511-2516, doi:10.3892/etm.2017.4793 (2017).
We included the full name for MSH in Abstract (Page 1, Lines 24), and corrected in all text the appropriate abbreviation for α-MSH. The changes are in red in the new version of the manuscript.

Reviewer 2 Report
This manuscript is very well written with appropriate Tables and Figures to support the Results.
The Methods are adequately described though incomplete (see below).
The Discussion is appropriate.
Comments.
Ln 2I would suggest rewording the first sentence as I do not think it is true. I believe dry eye syndrome by far is the more common than ocular allergic disease. Same situation in ln 41.
Ln 23 suggest spell out Tregs and thereafter use the abbreviation.
Ln 91 Table 1. Add description of abbreviation SPT (mm) to Der p in caption.
Ln 91 Table 1. Add definition of (sg) with TBUT
Note that in general most of the captions throughout the manuscript must be expanded upon to describe the abbreviations appropriately.
Note that there are a number of sections missing from the Material and Methods e.g. description of the determination of the TBUT; description of the determination of Schirmer's test results. The type of Schirmer analysis with or without anesthesia and why. The authors must describe why these measurements are important with regards to the manuscript.
Ln 337 Add City and State of source of solution.
Ln 401 Provide City and State of GraphPad Prism.
Ref 23 Capitalize "Network"
Ref 28-34 Remove italics from journal titles and vol #s where applicable to be consistent with the other references.
Ref 31 Capitalize "Reports"
Author Response
This manuscript is very well written with appropriate Tables and Figures to support the Results.
The Methods are adequately described though incomplete (see below).
The Discussion is appropriate.
Comments.
Ln 2I would suggest rewording the first sentence as I do not think it is true. I believe dry eye syndrome by far is the more common than ocular allergic disease. Same situation in ln 41.
We adjusted the abstract and introduction according to reviewer suggestion. Page 1, Lines 21, and line 41
Ln 23 suggest spell out Tregs and thereafter use the abbreviation.
Ln 91 Table 1. Add description of abbreviation SPT (mm) to Der p in caption.
Ln 91 Table 1. Add definition of (sg) with TBUT
Note that in general most of the captions throughout the manuscript must be expanded upon to describe the abbreviations appropriately.
We included Tregs' meaning and added description for abbreviation, (Page 1, line 23). We corrected abbreviature for the measurement unit in TBUT (Table 1, Page 3, line 99). We corrected abbreviation for SPT (Table 1, Page 3), and reviewed all manuscript carefully to describe abbreviations appropriately.
Note that there are a number of sections missing from the Material and Methods e.g. description of the determination of the TBUT; description of the determination of Schirmer's test results. The type of Schirmer analysis with or without anesthesia and why. The authors must describe why these measurements are important with regards to the manuscript.
In the new version of the manuscript, we included a description of the clinical evaluation, (Page 10, lines 339-355). We also reviewed the importance of Schirmer's test and TBUT measurements according to our results in Discussion, (Page 9, lines 305-322) and the reference section was actualized to include the new bibliography added to the text, all changes are in red.
Ln 337 Add City and State of source of solution.
Ln 401 Provide City and State of GraphPad Prism.
Ref 23 Capitalize "Network"
Ref 28-34 Remove italics from journal titles and vol #s where applicable to be consistent with the other references.
Ref 31 Capitalize "Reports"
All reviewer suggestions were performed and in the new version of the manuscript all changes are in red.

Round 2
Reviewer 1 Report
Authors have performed new experiment for α-MSH dose-response and answered my most comments properly. The quality of the revised manuscript is much improved although some English typo errors need to be corrected, such as a-MSH should be α-MSH.